# Deciphering the Potential Coding of Human Cytomegalovirus: New Predicted Transmembrane Proteome

**DOI:** 10.3390/ijms23052768

**Published:** 2022-03-02

**Authors:** Francisco J. Mancebo, Marcos Parras-Moltó, Estéfani García-Ríos, Pilar Pérez-Romero

**Affiliations:** 1National Center for Microbiology, Instituto de Salud Carlos III Majadahonda, 28222 Majadahonda, Spain; fj.mancebo@isciii.es (F.J.M.); mpperez@isciii.es (P.P.-R.); 2Department of Mathematical Sciences, Chalmers University of Technology, 41296 Gothenburg, Sweden; mparmol@gmail.com; 3Centre for Antibiotic Resistance Research (CARe), University of Gothenburg, 41122 Gothenburg, Sweden; 4Igenomix Foundation Research Department—INCLIVA, 46980 Valencia, Spain; 5Department of Science, Universidad Internacional de Valencia—VIU, Calle Pintor Sorolla 21, 46002 Valencia, Spain

**Keywords:** cytomegalovirus, pangenome, proteome, transmembrane

## Abstract

CMV is a major cause of morbidity and mortality in immunocompromised individuals that will benefit from the availability of a vaccine. Despite the efforts made during the last decade, no CMV vaccine is available. An ideal CMV vaccine should elicit a broad immune response against multiple viral antigens including proteins involved in virus-cell interaction and entry. However, the therapeutic use of neutralizing antibodies targeting glycoproteins involved in viral entry achieved only partial protection against infection. In this scenario, a better understanding of the CMV proteome potentially involved in viral entry may provide novel candidates to include in new potential vaccine design. In this study, we aimed to explore the CMV genome to identify proteins with putative transmembrane domains to identify new potential viral envelope proteins. We have performed in silico analysis using the genome sequences of nine different CMV strains to predict the transmembrane domains of the encoded proteins. We have identified 77 proteins with transmembrane domains, 39 of which were present in all the strains and were highly conserved. Among the core proteins, 17 of them such as UL10, UL139 or US33A have no ascribed function and may be good candidates for further mechanistic studies.

## 1. Introduction

Human cytomegalovirus (CMV) is a large envelope worldwide prevalent betaherpesvirus, ranging from 45% to 100% in the general population based on socio-economic factors [1,2]. Although CMV generally causes asymptomatic infections in immunocompetent individuals, it is a major cause of morbidity and mortality in immunocompromised individuals such as organ transplant recipients, AIDS, and with congenital infection [3,4,5,6,7].

The CMV genome (236 kb) consists of a unique long (UL) and a unique short (US) region flanked by inverted repeats. CMV gene expression occurs with the expression of immediate-early genes followed by early, early-late and late transcripts [8]. In addition to the 165 canonical ORFs [9,10], CMV genome encodes for other alternative transcripts in addition to have non-canonical translation initiation sites [11,12]. Furthermore, CMV encodes for a large number of genes, many of them with unknown functions that may probably be involved in key processes during host-cell interaction [13].

CMV is able to infect a high number of cell types including fibroblasts, endothelial cells, epithelial cells and myeloid lineage cells, among others [14,15]. CMV is a highly complex virus with multiple proteins embedded in the viral envelope, with at least four distinct types of covalently linked glycoprotein complexes required for CMV infectivity including gCI complex (gB dimer), gCII complex (gM, gN), gCIII complex (trimer gH, gL, gO) and pentameric complex (gH/gL/UL128-131) [16,17]. The therapeutic use of neutralizing antibodies, targeting glycoproteins mediating viral entry, have demonstrated to only achieve partial protection against CMV infection [18,19,20,21,22,23]. One possible explanation if that other proteins may be involved in viral entry that might be also necessary to target in order to elicit a complete protection against infection. An ideal vaccine against CMV infection should elicit a broad immune response, including both neutralizing antibody and T-cell response, against multiple viral antigens including proteins involved in virus-cell interaction and entry [24,25], which may increase efficacy compared with the previously tested vaccines [1,26,27,28]. In fact, despite the efforts made during the last decade, no CMV vaccine is still available [18,27,29,30]. Thus, understanding the complete repertoire of CMV proteins involved in cell entry may also help to determine the neutralizing response necessary to block infection and may provide novel candidates that could be included in new vaccines design.

In this study, we aimed to explore the CMV genome to identify proteins with putative transmembrane domains to identify new potential viral envelope proteins that may be involved in virus-cell interaction during infection and may therefore be potential targets for neutralizing antibodies for the development of novel therapeutics and preventive measures targeting viral entry and cell-to-cell spread. In order to do that, we have performed in silico analysis using the genome sequence of nine different CMV strains, to identify proteins with predicted transmembrane domains. To further characterize the identified proteins, an exhaustive systematic review of the literature and a sequence homology analysis with known proteins from other organisms were performed. Our work highlights the need to explore new experimental and computational approaches to identify and characterize the CMV proteome.

## 2. Results

### 2.1. Identification of Putative Transmembrane Proteins

To determine the transmembrane regions of the proteins encoded by the CMV genome, the genome of nine different CMV strains including both clinical and laboratory strains (Table 1) were analyzed using three different bioinformatic methods: Phobius, PureseqTM and TMHMM. A description of the methodological approach used is represented in Figure 1. CMV is known to accumulate mutations quite rapidly in cell culture during cell passaging [31]. In order to test to what level these nine selected CMV strains are representative of the 335 available CMV genomes in GenBank, the 56190 ORFs were aligned with the ORFs in our CMV dataset. We obtained 100% median percentage identity and breadth coverage (overlapping distance), representing 99.95% of the total ORFs from the Human betaherpesvirus 5 in the NCBI database (Appendix A).

Based on the first analysis, we identified 94 proteins with potential transmembrane domains (Figure 2 and Appendix A). Seventeen of them were not considered for further analysis (Appendix A) because of the following reasons. Proteins UL74, UL115, UL47, UL49, UL76, US22, UL77, UL105, UL122 and UL89 were previously described not to be part of membrane structures [32,33,34,35,36,37,38,39,40,41]. UL47, UL49, UL76 and US22 proteins are known to be part of the tegument, UL77 is located in the capsid; UL105, UL122 and UL89 are found in the nucleus of the host cells [32,33,34,35,36,37,38,39]. In addition, UL4, UL22A and UL116, which were predicted to have one transmembrane domain, were discarded because the transmembrane domain corresponded to the sequence of signal peptide [42,43,44]. In addition, RL13TRL14 and US33 (TB40-E_UNC strain) or ORFL27C and ORFL49W.IORF1 (AD169-BAC20) proteins were only found in one of the CMV studied strains and were not considered for further analysis.

For further characterization of the 77 remaining proteins with predicted transmembrane (TM) domains, a systematic review was performed to search for any previous published information. For each of the proteins, information on the location, the ascribed function and the number of predicted TM domains is included in Appendix A. A graphical representation of the number of predicted TM regions found for each ORF, in each of the nine strains with the indicated bioinformatics tool is shown in Figure 2. Of the 77 proteins analyzed, 33 (43%) proteins only had one TM domain, 23 (29.87%) had from 1 to 2 TM domains, 6 (7.7%) exhibited 1-3, while 15 proteins (19.48%) had from 5 to 8 TM domains. None of the analyzed proteins had four TM domains.

Nineteen out of the 77 proteins (UL2, UL6, UL9, UL14, UL15A, UL74A, UL120, UL121, UL140, UL148C, UL148D, US13, US15, US19, US29, US30, US33A, RL8A, RL9A and RL10) have no previously described function, 13 (UL1, UL5, UL8, UL10, UL20, UL42, UL78, UL124, UL139, UL147, US34A, RL12 and RL13) have been partially studied, 1 (UL41A) has previously been shown not to code for a protein [10] and the other 43 proteins have a previously described function (Table 2).

The number of predicted domains differed in some of the studied proteins when using different methods. The results obtained TMHMM method were the most divergent of the three tested methods. On the contrary, a group of proteins encoded by the genes UL33, UL78 and the genes from the unique short (US) region US12-US21, US27 and US28 proteins were predicted to have more than five TM regions by all three methods. In fact, TM regions of these genes, such as the members of US12 family and the proteins with homology to the chemokine receptor family of G protein-coupled receptors (GPCRs): US27 and US28, have been previously described supporting our results [108,109,121].

A validation experiment was carried out using as an example UL2 and UL124, two of the identified proteins with unknown function. The ORF encoding for these two proteins were cloned into a eukaryotic expression plasmid that included a Myc tag sequence in the 5´end of the clone products. After transfecting the HEK 293T mammalian cell line, plasma membrane proteins were extracted and the cytoplasmic (C) and plasma membrane (PM) protein fractions were tested by Western Blot using an anti Myc antibody. As shown in Appendix A, both UL4 and UL124 proteins were only detected in the PM fractions confirming their location in the membrane. A loading control using the stain free technology is shown in Appendix A.

### 2.2. Homology Analysis of thePredicted Transmembrane Proteins

In addition to the exhaustive systematic review of the literature, further analysis of sequence homology with known proteins from other organisms was performed using Mantis software (Appendix A). Based on this analysis, we found homologies for two of the proteins with unknown function. UL139 had some level of homology (*e*-value = 5.1 × 10^−28^) with proteins involved in cell adhesion, while UL15A had some homology (*e* value = 1.53 × 10^−4^) with a biotin permease protein. UL15A ORF was identified in all 9 CMV strains analyzed, while UL139 that was only present in the TR strain.

In addition, Mantis analysis shed an association of UL1 with a carcinoembryonic antigen-related cell adhesion molecule, which is a cell adhesion receptor of the immunoglobulin-like superfamily. UL78 was also identified by Mantis as seven transmembrane receptor from the rhodopsin family. UL147 has been proposed by Mantis to be involved in immune response and chemokine activity and US33A seems to have a von Willebrand A (VWA) domain. However, US33A was present exclusively in Towne, Toledo, TR and VR7863 strains.

### 2.3. Sequence Differences among Strains

The analysis of the generated pangenome (Figure 2 and Figure 3), revealed wide differences among strains [13]. Clinical isolate VR7863 and TB40-E_UNC strains lacked a large number of genes compared to the other strains. The VR7863 strain lacked some genes with unknown functions such as UL1, UL6, UL139, UL140, US13, US15, US19 and US29 and other genes involved in immunoevasion, DNA packaging, latency or viral replication such as UL37, UL40, UL119, UL133, UL135, UL148A and US20 [39,60,61,62,73,76,77,93,106]. The TB40-E_UNC strain lacked some genes with unknown function such as UL6, UL74A, UL140, US12, US19, US29, US33A, RL8A, RL9A and RL13 and other genes involved in host immune response evasion, CMV assembly, tropism, or latency such as UL119, UL132, UL133, US2, US3 and US16 [73,75,95,96,102,104,118,119].

AD169 strain have a deletion of a genomic region that included UL140, UL141, UL142, UL144 genes and RL13 gene (known to have TM domains) [117]. All of them have functions related to the evasion of the host’s immune system [84,85,86,87,88]. In addition, the AD169 BACmid (Table 1) widely used for research, lacked several genes encoding TM proteins [117]. Some of them such as UL140, UL141, UL142, UL144 and RL13 were also deleted in AD169 strain. While the AD169 BACmid also lacked other genes such as UL135, UL136, UL138, UL148 and US3-US6 genes, involved in DNA replication, latency, virulence, tropism, evasion of the immune response and other genes such as UL139, UL147, UL148B, UL148C and UL148D with uncharacterized function [77,79,81,92,96,97,117]. The Toledo strain lacked RL13, UL9 and UL128 genes, while the Towne strain lacked RL13, UL1 and UL40 genes. The TR and Merlin strains, widely used in research included all the analyzed genes.

### 2.4. Sequence Similarities among Strains: Core Proteins

Based on the differences among different strains, we searched for those genes with predicted TM domains that were present in all the studied CMV genomes. Of the 77 initially predicted TM proteins, 39 (50.64%) met the criteria and were designated as the core TM proteome (Figure 2 and Figure 3). A representation of the 39 proteins grouped according to their function is shown in Figure 3A. The number of predicted TM domains is also indicated in each group. No association was found between the number of TM domains and the function in each group.

Most of the 39 core proteins were highly conserved among all nine strains with a high percentage of sequence similarity (94.33 ± 7.3), except for the RL12 gene (48.31 ± 24.17, Figure 4 and Appendix A). The similarity matrix for each protein in each indicated strain is shown in Appendix A and an example of heat map depicting the similarity of the core proteins comparing all the strains with AD169, as a reference strain, is shown in Figure 4. As expected, AD169-derived BAC was almost identical to the AD169 strain. Despite the fact that sequence similarity was overall high (with an average above 94%), the sequences of the core proteins from the Merlin, Towne and TR strains differed the most. In addition, most of the core proteins with unknown function (highlighted in red in Figure 4), tended to have lower similarity values compared with other core proteins with known functions.

Some of the 39 identified core proteins were involved in cell entry (4, 9.52%) and immunomodulation/immunoevasion mechanisms (13, 33.33%). However, a significant percentage of them (17, 43.58%) had no described function (UL2, UL8, UL14, UL15A, UL41A, ULL120, UL121, UL124, US30, RL10 and RL12) or were poorly studied (UL5, UL10, UL20, UL42, UL78 and US34A). Figure 3B shows the location for each of the 18 core proteins with uncharacterized function in the CMV genome.

In addition, in order to test to what extent the core proteins were found among the CMV population, the 335 available genomes in the database were aligned to our set using blastp to all proteins present in our pangenome core. All genomes had representative proteins related to proteins in our set in different proportions with a high number of genomes containing all 39 core proteins, indicating that our pangenome could be extended to all annotated genomes (Appendix A).

### 2.5. CMV Gene Families with Transmembrane Domains

The genes encoded by the CMV genome are grouped into several gene families [122], five of which are important families that include genes with TM domains. Thus, we next analyzed which of the 39 identified core proteins are part of these families.

The RL11 family share the RL11D central domain that includes three conserved residues (one tryptophan and two cysteines) and several potential N-linked glycosylation sites associated with immunomodulatory properties. The RL11 family consists of 14 genes (RL5A, RL6, RL11-13, UL1 and UL4-11 genes), encoding for proteins with transmembrane domains, except for RL5A and RL6 [51], most of them with unknown specific function [123]. Based on our analysis, we identified three of the genes belonging to this family (UL5, UL8 and UL10) that have not been well characterized, although they are suggested to be involved in the viral cycle and immunomodulation mechanisms (Appendix A) [42,45,52].

The US12 family includes US12-US21 genes [102]. US12, US14, US18 and US20 had been shown to be involved in the inhibition of natural killer cells [107]. US16 had been shown to interact with UL130 participating in CMV tropism [104]. US21 encodes a viroporin that modulates calcium homeostasis and protects the cells against apoptosis [108]. US18 and US20 were described to play a role in cell tropism mediating viral replication in specific cells while US13, US15 and US19 have no described function or are poorly characterized [102,107].

The GPCR-7 TM family includes four genes coding for proteins with seven predicted TM domains: US27, US28, UL33 and UL78. This family includes G protein-coupled surface receptors with an important role in immunomodulation that transmit an intracellular signal when binding to an extracellular ligand [109,124]. Within this family only UL78 (protein from the unknown core proposed in this work) remains uncharacterized.

UL120 and UL121 are proposed to form a family and both of them were identified in our unknown core analysis. However, little is known about their function and further studies are needed to shed light into their biological relevance [125].

## 3. Discussion

The lack of knowledge of an important part of the genes encoded by the CMV genome, the variability between laboratory and clinical strains and the complex regulation of the virus over the host immune system, have probably limited the design of new preventive and prophylactic measures against CMV infection [1,122,126]. CMV proteins involved in the interaction with the target cells during entry located in the viral envelope may be considered possible targets to develop new treatments and vaccines against CMV since blocking a combination of these proteins may block the infection in the different target cells [24]. However, of the 165 proteins potentially encoded by the CMV genome, only around 60 proteins have been functionally characterized [12,112,127,128]. In this sense, some authors have reported the association between γ marker, human leucocyte antigens and killer immunoglobulin-like receptors and the natural course of CMV infection [129,130,131,132].

We performed the present study using in silico analysis of the genomic sequences of nine CMV strains (representative of the 335 available CMV genomes in GenBank) with three different bioinformatic tools, to identify proteins that have putative TM domains, to identify new potential envelop proteins, that may help to better understand CMV interaction with the cells and with the host immune system.

As a result, we gained knowledge about the performance of the three bioinformatics tools used in this study. While the Phobius and TMHMM tools are well-established methods to study TM regions (using sequence-based approaches that use hidden Markov), PureseqTM is a novel alternative based on a machine learning model algorithm (DeepCNF), which has demonstrated to increase the number of results obtained. These tools have already been used to study membrane proteins in other organisms such as humans, *Plasmodium falciparum*, or *E. coli* [133,134,135]. Although these tools are quick in silico approaches to analyze the genome of a given organism, some of the results obtained are not always accurate. We obtained some variability of the number of predicted TM domain as a function of the used method. These discordances between methods could be explained due to differences in the algorithm and the threshold values. By default, each applied method that predicts TM regions uses their own algorithms and thresholds, probably detecting slight differences for the same region resulting in different thresholds. The analysis of the signal peptide was also important to exclude false positives since the signal peptides of type I TM proteins are usually hydrophobic and are often predicted to be TM domains. In addition, the nucleotide sequence variability of the different CMV strains with a significant number of polymorphisms (such as genes gN, and UL21) may also explain this variability [56,136,137]. These results highlight the importance of applying different bioinformatics methods for predicting domains in silico.

Proteins that are evolutionarily related are commonly referred to as homologues and very close homologues often have similar functions [138]. Homology-based methods have been previously used in different scenarios such as the identification of oncogenes from retrovirus proteins; cancer metastasis; herpesviruses; and *Hepadnaviridae* [139] among others. Using a homology-based method, we proposed functions for 23 CMV proteins that were previously characterized, which confirmed the performance of the method. We were also able to propose putative functions to several unknown CMV proteins such as UL1, UL15A, UL139, UL78, UL147, US13 and US33A. UL15A was proposed as a biotin transport system permease protein, while UL139 could be involved in cell adhesion. Furthermore, based on the homologies, UL1 was identified to be involved in the viral-cell adhesion and potentially modulating the immune system of the host. Most of those carcinoembryonic antigen-related cell adhesion molecules are modulators of general cellular processes such as proliferation, motility, apoptosis as well as cell-cell interaction that binds to pathogens enhancing their capacity to colonize the host [140,141]. Our analysis identified UL78 as a member of the rhodopsin family which is large group of evolutionarily-related proteins that are cell surface receptors, detecting molecules outside the cell and activate cellular responses [142]. However, given the functional homology among the other members of the family, it is plausible to think that UL78 exhibits similar functions and may have an immunomodulatory role [113], although its ligand is still unknown. UL147 arises with a potential role in immune response and chemokine activity, likely due to its homology with UL146 which is already characterized [90,143]. Finally, US33A, which is present in Towne, Toledo, TR and VR strains, showed a von Willebrand A (VWA) domain that is well characterized to be involved in cell adhesion with extracellular matrix proteins and integrin receptors [144] and could be likely be involved in signaling.

## 4. Materials and Methods

### 4.1. Transmembrane Region Analysis

The nine CMV genome sequences AD169, Towne, Toledo, TR, VR7863, TB40-E_UNC, HANSCTR4, AD169-BAC20 and Merlin (Table 1) were available at the Nucleotide database. In order to test to what level these nine selected CMV strains are representative of the available CMV genomes in GenBank and the ORF from the 335 available CMV genomes containing 56217 ORFs (annotated in the NCBI database as human betaherpesvirus 5 complete genomes) were downloaded. Sequences were aligned using blastp with default parameters. Additionally, the complete set of ORFs coding for CMV proteins was aligned with the ORFs of the protein core in the pangenome. Blast results were filtered for sequence percentage identity ≥75% and breadth coverage ≥90%.

These nine genome sequences were analyzed to predict transmembrane domains within the open reading frames (ORFs). The analysis was carried out using the default parameters of three different bioinformatics approaches: PureseqTM (v1.2) [145], Phobius (v1.01, Stockholm Bioinformatics Center, Sweden) [146] and TMHMM (DTU Health Tech, Lyngby, Denmark) (v1.1) [147]. Phobius and TMHMM are based on sequence methods as hidden Markov model (HMM) approach, while PureseqTM adds an extra layer of prediction based on deep learning. All methods were expected to show similar output with differences in the predictive threshold for the same set of analyzed ORFs. The tool SignalP6.0 (DTU Health Tech, Lyngby, Denmark) was used for signal peptide analysis [148].

### 4.2. Functional Annotation of the Transmembrane ORFs

Predicted TM proteins were analyzed using Mantis protein function annotation v1.1.1 [149] which is a stand-alone tool that uses HMMER or Diamond to match sequences against multiple reference datasets to produce high-quality consensus driven protein annotations, under default parameters. This analysis uses the information from the following different available protein function databases: KOfam [150], Pfam [151], eggnog [152], NCBI protein family models (NPFM) [153], and TIGRfams [154] and sets a consensus result. In parallel, we searched for sequence homology in the RefSeq database using blast with Blastp (v2.9.0) [155] (*e*-value < 10^−3^) looking for orthologous proteins. Orthologous proteins are proteins found in other species that maintain the same or close functionality as the studied protein. Proteins with unknown functions could be related to their orthologous protein function this way.

Additional systematic review was performed to retrieve related articles published on the PubMed database website (https://pubmed.ncbi.nlm.nih.gov/, accessed on 15 December 2021) until July 2021. For each of the genes studied, articles were identified using the following search terms: “HHV-5” AND or “CMV”.

### 4.3. Viral Pangenome Construction

A pangenome was created for all nine sequences with Roary (v3.11.2, Wellcome Genome Campus, Hinxton, UK) [156] under default parameters. Pangenome representation and TM information from all genomes were plotted using ggplot2 (v3.3.3) [157] and ggcorrplot (v0.1.3) R packages.

### 4.4. Similarity Analysis

Core proteins for all nine CMV strains were aligned using Clustal Omega (v1.2.1) [158] with -percent-id and –full parameters to obtain identity matrices. Thirty-nine matrices were obtained and results were plotted as individual heatmaps using heatmap.2 from gplots package (v3.1.1). In parallel, a percentage identity heatmap was plotted for the comparison between AD169 and the other studied strains 39 core proteins, following the same parameters.

### 4.5. Protein Analysis

UL2 and UL124 ORFs were amplified by PCR using CMV BADrUL131-Y4 BAC as a template, with specific primers (Appendix A), and the Phusion DNA polymerase (Thermo Scientific). PCR products and the pcDNATM3.1/myc-His (-) (5.5 kb) vector (Invitrogen) were digested with the appropriate restriction enzyme (FastDigest enzymes, Thermo Scientific, Waltham, MA, USA), ligated (Ligase, Thermo Scientific, Waltham, MA, USA) and transformed into the XL10 Gold chemically competent *E. coli* cells. The constructed plasmids were verified by sequencing (Appendix A). The pcDNATM3.1/myc-His (Thermo Scientific, Waltham, MA, USA) constructs containing the UL2 and UL124 ORFs were transfected into the HEK 293T human cell line, using the CaCl_2_ transfection method. Briefly, the day before transfection, 1,500,000 HEK 293T cells were seeded in a 10-cm plate and the next day transfections were carried out with approximately 70–80% of cell confluence. Four hours before the transfection, fresh medium was added to cells. For the transfection, 750 μL of CaCl_2_ were mixed with 40 μg of the DNA construct. 750 μL of HBS saline buffer (140 mM NaCl, 1.5 mM Na_2_HPO_4_, 50 mM HEPES) were added and the mixture was incubated for 15 min at room temperature. Subsequently, the mixture was added dropwise to each plate containing the 293T cell monolayer, and gently mixed. Transfected cells were incubated during 48 h at 37 °C with 5% CO_2_. Twenty-four hours post-transfection, fresh medium was added to cells. Transfected cells were pelleted and treated with Mem-PER^TM^Plus Membrane Protein Extraction Kit (Thermo Scientific, Waltham, MA, USA) according to the manufacturer’s instructions. Cytoplasmic and plasma membrane protein fractions were obtained and quantified by Bradford. Ten µg of protein lysates were separated on a gradient 4–20% pre-cast SDS gel (BioRad, Hercules, CA, USA), transferred to 0.2 µm nitrocellulose membranes. Detection of the expressed proteins was performed using an anti-Myc monoclonal antibody (MA1-21316, Invitrogen, Waltham, MA, USA) at a 1:1000 dilution in blocking buffer (1X PBS + 0.1% Tween 20 + 5% skim milk) and incubated at 4 °C overnight, with a secondary horseradish peroxidase (HRP)-labeled anti-mouse IgG antibody (diluted 1:2000; 05/2019, Cell Signaling, Danvers, MA, USA). Stain free technology (BioRad, Hercules, CA, USA) of the acrylamide gel was used in parallel as a loading control.

## 5. Conclusions

Our results highlight the utility of using these bioinformatic tools to gain knowledge of previously uncharacterized proteins that may be useful to select potential targets of the immune system. Our work also suggests that differences among the strains may be crucial for CMV tropism, replication, latency or the evasion of the host immune response [13]. Among the 77 identified proteins with predicted TM domains, only 39 (designated as a core TM proteome) were shared by all analyzed strains most of which were highly conserved which may have potential clinical relevance for the design of new therapeutics and preventives measures. This group of proteins was highly conserved in all nine strains analyzed, which may have solved the limitation of the variability among laboratory strains and clinical isolates, facilitating the extrapolation of the results.

In addition, the study of the role of the previously uncharacterized proteins may provide novel candidates for new preventive or prophylactic measures against CMV infection. Furthermore, if their location in the viral envelope is confirmed, they could be used as viral antigens for developing a vaccine or monoclonal antibodies. It is noteworthy to highlight protein candidates such as UL139 and US33A, which seem to be involved in the adhesion with the host cell or UL10 in which its location in the membrane has been demonstrated but its function has not been fully characterized [52].

In conclusion, using a complex in silico analysis we have predicted CMV proteins with TM domains that could be of interest because of their possible role in virus-cell interaction and entry. Our approach has been very useful to search for new potential candidates for a more rational design of new treatment targets and vaccines as well as to increase our knowledge of CMV.

## Figures and Tables

**Figure 1 ijms-23-02768-f001:**
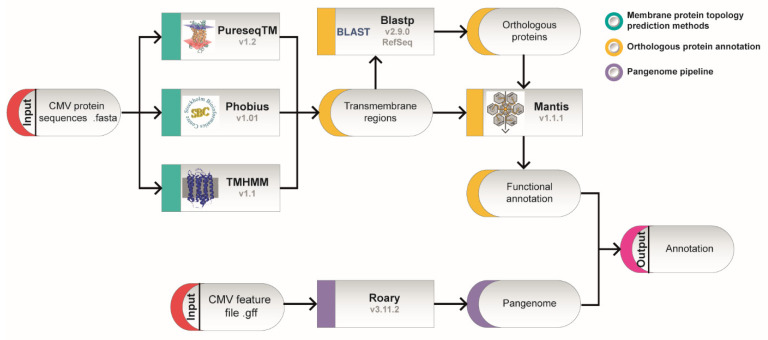
Schematic representation of the applied workflow. Fasta format protein sequences from nine CMV genomes were analyzed in parallel to predict transmembrane domains and to create an entire set of genes from all strains (pangenome). Transmembrane topology was studied following three different approaches: PureseqTM, Phobius and TMHMM, under default parameters. Predicted transmembrane proteins were compared with orthologous proteins identified by BLAST with the whole Mantis database for the prediction of functional annotation. Proteins that were common to all nine genome datasets formed the core protein set, and functions were annotated accordingly for each transmembrane protein.

**Figure 2 ijms-23-02768-f002:**
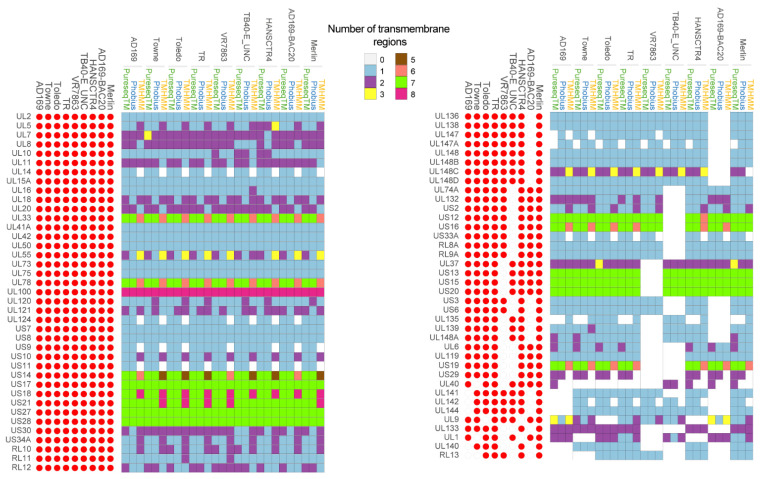
Predicted transmembrane proteins for the studied CMV genomes. Proteins with at least one predicted transmembrane domain with one of three tested methods were annotated as transmembrane proteins. The number of transmembrane domains for each protein was represented using the indicated chromatic scale ranging from zero to eight regions for each of the three methods used (PureseqTM, Phobius and TMHMM). For each strain, the presence of the gene was represented with the filled red circles and absent genes with empty red circles.

**Figure 3 ijms-23-02768-f003:**
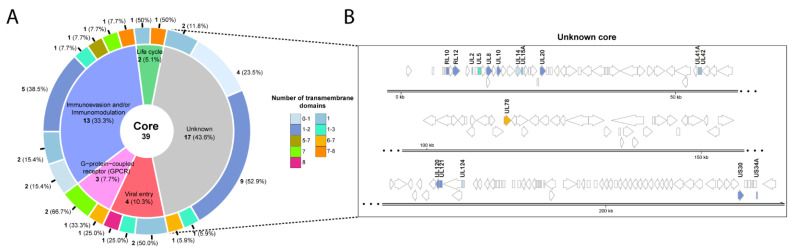
Functional analysis of the 39 core proteins. (**A**) Pie chart of the proteins found in all the studied strains were grouped based on their functions. For each group, the number of predicted transmembrane domains is also indicated. When the number of transmembrane domains predicted was different using the three methods, a range of values is shown. The number of proteins in each section is marked in blue and the percentage between brackets. (**B**) Genomic location of the 17 proteins with non-described function.

**Figure 4 ijms-23-02768-f004:**
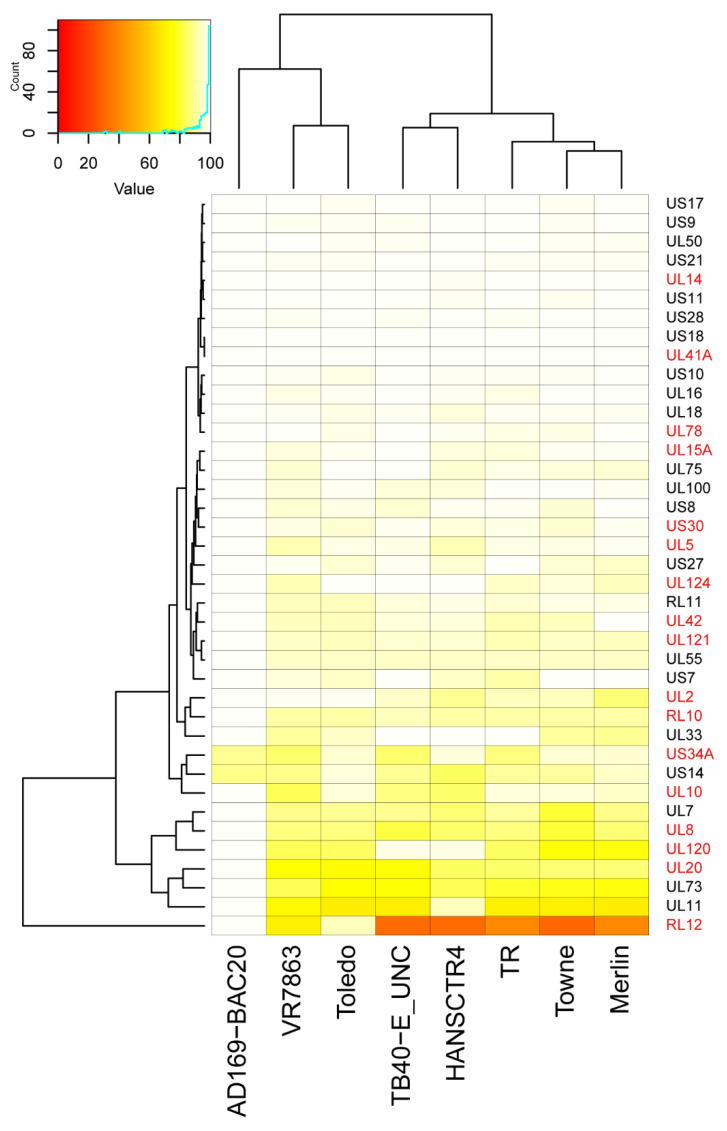
Percentage identity heatmap among the 39 core proteins from all the studied strains using AD169 as a reference strain. Color scales range from red (0% identity) to white (100% identity). Strains and genes were clustered following a hierarchical clustering method (HCL). Genes were clustered from top to bottom of the figure based on their similarity within the indicated strains. Core proteins with unknown functions are highlighted in red.

**Table 1 ijms-23-02768-t001:** Characteristics of CMV strains used in this study and their corresponding accession number at nucleotide database.

CMV Strains	Isolation Source	Number of Culture Passages	Accession Number
AD169	Adenoids of a 7-year-old girl	Many times in human fibroblasts	FJ527563.1
Towne	Urine of a 2-month-old infant with microcephaly and hepatosplenomegaly	Many times in human fibroblasts	FJ616285.1
Toledo	Urine from a congenitally infected infant	Several times in human fibroblasts	GU937742.2
TR	Vitreous humor from eye of HIV-positive male	Several times in human fibroblasts	KF021605.1
VR7863	Urine samples of a congenitally infected neonate and cultured in endothelial and epithelial cells	Cultured in endothelial and epithelial cells	KX544838.1
TB40-E_UNC	Throat swab of a bone marrow transplant patient	Cultured adapted	KX544839.1
HANSCTR4	Blood from stem cell transplant recipient (D-R+)	Sequenced directly from clinical material via target enrichment	KY123653.1
AD169-BAC20	-	-	MN920393.1
Merlin	Urine from a congenitally infected child	3 times in human fibroblasts	NC006273.2

**Table 2 ijms-23-02768-t002:** CMV predicted transmembrane proteins indicating the cellular localization based on biotool Uniprop, the ascribed functions based on a bibliographic search and the number of predicted domains using the three different tools. (*) indicates unknown or non-verified function.

Gene	Localization	Function	Number of TM Domains	References
UL1 *	VM	Unknown. pUL1 could modulate CMV host cell tropism.	1–2	[45]
UL2 *	HM	Unknown.	1	-
UL5 *	V	Unknown. It is suggested to be involved in efficient viral assembly, propagation and replication.	1–3	[46,47]
UL6 *	HM	Unknown.	1–2	-
UL7		UL7 is involved in immunomodulation.	2–3	[48,49,50]
UL8 *	HM	UL8 decreases the release of a large number of pro-inflammatory factors later after infection of THP-1 myeloid cells. UL8 may exert an immunosuppressive role key for CMV survival in the host.	1–2	[51]
UL9 *	HM	Unknown function. Its deletion mutation cause enhanced growth in HFFs cells.	1–3	[9]
UL10 *	M	Unknown. Potential role in immunomodulation.	1–2	[52]
UL11	HM, ERM	pUL11 interacts with CD45 phosphatase on T cells, inducing the IL-10 secretion.	1–2	[53]
UL14 *	HM	Unknown.	0–1	-
UL15A *	HM	Unknown.	1	-
UL16	HM	Immunoevasion and inhibition of the activation of NK cells.	1	[54]
UL18	HM	Immunomodulation and immunoevasion.	1–2	[55]
UL20 *	ERM	Unknown. UL20 could be destined to sequester cellular proteinases not known to date for degradation in lysosomes.	1–2	[56]
UL33	HM	UL33 has homology with GPCR which activates different ligand-independent signalling pathways and also involved in virus dissemination.	6–7	[57,58,59]
UL37	ERM, GM, MM	Viral replication.	2–3	[60,61]
UL40	HM	Immunomodulation.	0–2	[62]
UL41A *	VM	Unknown. UL41A not to code for proteins.	1	[10]
UL42 *	HM, C	Unknown. Potential role in immunoevasion.	1	[63,64]
UL50	HNM	Assembly, maturation and egress of virions.	1	[65]
UL55	VM, HM, GM	Glycoprotein B participates in viral entry.	1–3	[66]
UL73	VM, HM, GM	Glycoprotein N is involved in the binding of the virus to the host cell, viral spread and virion morphogenesis.	1	[67]
UL74A *	VM	Unknown	1	-
UL75	HM, VM	Glycoprotein H participates in viral entry. It is part of the trimeric and pentameric complexes.	1	[68]
UL78 *	HM, ERM	Unknown. UL78 is a G protein-coupled receptor.	6–7	[69,70]
UL100	HM, VM	Envelope glycoprotein M participates in viral entry.	8	[71,72]
UL119	VM	Immunoevasion.	1	[73]
UL120 *	HM	Unknown.	1–2	-
UL121 *	HM	Unknown.	1–2	-
UL124 *	HM	Potential role in latency.	0–1	[74]
UL132	VM	Essential for CMV assembly compartment formation and the efficient production of infectious particles.	1–2	[75]
UL133	GM	UL133 forms a complex with UL138 and UL136. It is involved in the establishment of CMV latency.	2	[76]
UL135	HM, GM	Immunomodulation. Post entry Tropism in Endothelial Cells.	0–1	[77,78]
UL136	HM	Replication, latency, and dissemination. Post entry Tropism in Endothelial Cells.	1	[76,78,79,80]
UL138	GM	Latency and DNA replication.	1	[81,82]
UL139 *	HM	Unknown. Potential role in immunomodulation.	1–2	[83]
UL140 *	HM	Unknown.	1	-
UL141	ERM	Immunomodulation and DNA replication.	1	[84,85,86]
UL142	ERM	Immunomodulation.	0–1	[87]
UL144	HM	Inhibition of T-cell activation and latency.	1	[88,89]
UL147 *	EXR	Unknown. Potential role in immunomodulation.	0–1	[90]
UL147A	HM	Immunomodulation.	0–1	[91]
UL148	ERM	Viral ER-resident glycoprotein that interacts with UL116 promoting the incorporation of gH/gL complexes into virions.	1	[92]
UL148A	HM	Immunoevasion of NK cells.	1–2	[93]
UL148B *	HM	Unknown.	1	[94]
UL148C *	HM	Unknown.	0–3	[94]
UL148D *	HM	Unknown.	1	[94]
US2	ERM	Immunomodulation.	1–2	[95]
US3	ERM	Immunoevasion.	1	[96]
US6	ERM	Immunomodulation.	1	[97]
US7	ERM	Immunoevasion.	1	[98]
US8	ERM, GM	Immunomodulation.	1	[98]
US9	ERM, GM, CK	Glycoprotein US9 is an antagonist of IFN signalling to persistently evade host innate antiviral responses.	0–1	[99]
US10	ERM	Inhibition of the host immune response.	1–2	[100]
US11	ERM	Inhibition of the host immune response.	0–1	[101]
US12	HM	Inmunomodulation of NK cells activation.	6–7	[102]
US13 *	HM	Unknown.	7	-
US14	HM	Inmunomodulation of NK cells activation. Potential role in virions maturation and egress.	5–7	[102,103]
US15 *	HM	Unknown.	7	-
US16	HM, C	Tropism in endothelial and epithelial cells.	6–7	[104]
US17	HM	Immunomodulation.	7	[105]
US18	HM.	Immunoevasion of NK cell.	7–8	[106]
US19 *	HM	Unknown. Its delection affect NK cell activation.	6–7	[102]
US20	M	Inhibition NK cell activation. Also participates in the viral replication process in endothelial cells.	7	[106,107]
US21	HM	Viroporin that modulates calcium homeostasis and protects cells against apoptosis.	7–8	[108]
US27	V, HM	Immunomodulation. Also is required for efficient viral spread by the extracellular route.	7	[109,110,111]
US28	HM	Immunomodulation. Lytic and latent CMV infection. Possible role in regulation of the actin cytoskeleton or cytoskeletal remodelling.	7	[112,113]
US29 *	HM	Unknown.	0–2	-
US30 *	HM	Unknown.	1–2	-
US33A *	-	Unknown.	0–1	[114]
US34A *	HM.	Unknown. Potential target of SUMO complex.	1–2	[115]
RL8A *	HM	Unknown.	1	-
RL9A *	HM	Unknown.	1	-
RL10 *	VM	Unknown.	1–2	-
RL11	HM	Immunomodulation. RL11 is a type I transmembrane glycoproteins which bind immunoglobulin G Fc. I	1–2	[116]
RL12 *	VM	Unknown. RL12 is a Fc binding protein.	1–2	[117]
RL13 *	VM	Unknown. Potential role in replication, immunoevasión and viral spread by cell-free or cell-to-cell mechanisms.	1	[118,119,120]

* indicates unknown or non-verified function. CK: Cytoskeleton C: Cytoplasm, ERM: Host endoplasmic reticulum membrane, EXR: Extracellular region, GM: Golgi reticulum membrane, HM: host membrane, HMN: Host nucleus membrane, M: Membrane, MM: Mitochondrion membrane, V: Virion, VM: Virion membrane.

## Data Availability

The data presented in this study are available on request from the corresponding author.

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
