# Peer review of "Deciphering the Potential Coding of Human Cytomegalovirus: New Predicted Transmembrane Proteome"

_ijms, 2022, doi:10.3390/ijms23052768_

Round 1

Reviewer 1 Report

major:

Figure 1, 2, and 3 resolution is low

Figure 4 labelling in symbols?

Author Response

Dear Dr. Hornjak,

Thank you for your comments and for those of the reviewers regarding our recently submitted manuscript. We appreciate the time and effort that have gone into reviewing our manuscript and for the constructive comments of the reviewers.

We have modified the paper, taking into account the reviewers’ comments. A point-by-point response is included below.

We hope that the manuscript is now acceptable for publication.

We look forward to your response.

Sincerely,

Pilar Pérez-Romero

Response to reviewer:

REVIEWER 1

Figure 1, 2, and 3 resolution is low

Figure 4 labelling in symbols?

Response: We appreciate the reviewer’s comments and suggestions. Following the reviewer’s suggestion, we have improved the resolution and the symbols of the Figures.

Reviewer 2 Report

In IJMS manuscript number ijms-1563253, Mancebo et al. performed an in silico analysis for novel viral proteins containing putative transmembrane domains within the Cytomegalovirus genome of nine different strains of the virus.  Through sequence analysis, cross-referencing of the genomes of the nine strains, and comparing their findings to published literature, they obtain 42 genes with putative transmembrane domains.  19 of these had no previously described function.  These potential viral glycoproteins could be used by the virus for entry and for cell-to-cell spread.  They are potential therapeutic and vaccine targets for disruption of CMV infection.  The putative annotation of the 19 novel genes will be of significant benefit to CMV researchers interested in the natural history of CMV, its viral replication cycle, and CMV host-pathogen interactions.

The manuscript is clearly and concisely written.  The conclusions are supported by the results and the study will be of high impact to those interested in CMV.  I recommend acceptance for publication.

Author Response

Dear Dr. Hornjak,

Thank you for your comments and for those of the reviewers regarding our recently submitted manuscript. We appreciate the time and effort that have gone into reviewing our manuscript and for the constructive comments of the reviewers.

We have modified the paper, taking into account the reviewers’ comments. A point-by-point response is included below.

We hope that the manuscript is now acceptable for publication.

We look forward to your response.

Sincerely,

Pilar Pérez-Romero

REVIEWER 2

In IJMS manuscript number ijms-1563253, Mancebo et al. performed an in silico analysis for novel viral proteins containing putative transmembrane domains within the Cytomegalovirus genome of nine different strains of the virus.  Through sequence analysis, cross-referencing of the genomes of the nine strains, and comparing their findings to published literature, they obtain 42 genes with putative transmembrane domains.  19 of these had no previously described function.  These potential viral glycoproteins could be used by the virus for entry and for cell-to-cell spread.  They are potential therapeutic and vaccine targets for disruption of CMV infection.  The putative annotation of the 19 novel genes will be of significant benefit to CMV researchers interested in the natural history of CMV, its viral replication cycle, and CMV host-pathogen interactions.

The manuscript is clearly and concisely written.  The conclusions are supported by the results and the study will be of high impact to those interested in CMV.  I recommend acceptance for publication.

Response: Thank you for the positive input on the manuscript. We appreciate the reviewer’s comments.

Reviewer 3 Report

The authors of this manuscript did an in-silico analysis of 9 HCMV strains to identify viral proteins with transmembrane (TM) domains. They used the GenBank-annotated sequences of these 9 HCMV and analyzed them with three prediction software packages, PureseqTM, Phobius, and TMHMM. With these methods, 94 proteins were initially predicted to be putative TM proteins. 14 of those were excluded from further analysis because they have been shown not to be TM proteins in published experimental studies. 42 of the remaining 80 proteins were conserved in all 9 strains analyzed and were defined as core TM proteome. The proteins were further subdivided into different categories based on published data and homology to proteins from other organisms or pathogens.

Overall, the study is well presented and easy to read. The classification as a putative transmembrane protein may be helpful for future studies. However, the reader would benefit from information regarding the reliability of these predictions. Unfortunately, none of the proteins newly predicted to be TM proteins have been analyzed in biological experiments. Thus, it remains unclear how many of the predicted TM proteins truly are TM proteins – 14 were already discarded as obvious false positives.

Specific questions and comments

  1. The authors chose 9 HCMV strains (actually only 8 as AD169 is included twice) for their analyses, but they do not explain why they chose these strains. In fact, this reviewer would argue that the choice of HCMV strains is rather poor. 3 strains have been passaged extensively in cell culture (AD169, AD169-BAC20, Towne) or at least for several passages (Toledo, TB40E, TR). As HCMV is known to accumulate mutations quite rapidly in cell culture, the usefulness of these cell culture-adapted strains is rather limited (PMID 25894764). They encode numerous mutations, deletions, and sometimes even inversions, which will confound the analyses. Hence, the most suitable strains among the 9 are probably Merlin and HANSCTR4, maybe also VR7863. These strains most likely encode a true and complete HCMV proteome. There are more than 200 complete HCMV genomes in GenBank. In some cases, the viral genome sequence was determined directly from clinical material, prior to cell culture passaging. Why did the authors not use such HCMV genomes for their comparative analyses?
  2. Table 1 lists the 9 HCMV strains used in this study and the original sources of the strains. The information as presented is incomplete and therefore misleading. The authors should include in the table information on how extensively these viruses have been passaged in cell culture before the sequence was determined. They should also make the non-expert reader aware of the fact that the extensively passaged HCMV stains are not genetically intact.
  3. Dis the authors re-analyze / re-annotate the viral genomes or did they rely on the annotations found in the GenBank files? The ORF predictions are often based on simple algorithms. There are several proteomic studies that have used proteomic methods to analyze the protein coding content of HCMV. These studies have shown that some of the annotated ORFs do not code for proteins, whereas other previously not identified ORFs have been identified (e.g., PMID 23180859, 32486127, 31873071). Did the authors use information from these proteomic studies to curate their ORF database? One proteomic study has specifically focused on membrane proteins (PMID 24906157). Did the authors consider the results of this study?
  4. Defining a “core TM proteome” (Fig 3) based on a comparison of 9 strains, several of which are severely mutated, does not seem appropriate. The authors could use BLAST (or a similar software) to determine whether the 80 putative TM proteins are conserved in most HCMV strain sequences present in GenBank. Excluding a protein because it is not present in AD169 or Towne seems totally inappropriate.
  5. The signal peptides of type I TM proteins are usually quite hydrophobic and are often predicted to be TM domains , which they are not as the signal peptide is not present in the mature protein. Did the authors account for signal peptides in their analyses?

Minor points

  1. Figure 4. There are weird characters and symbols in this figure, probably due to use of an incompatible font.
  2. The supplemental figures are in need of legends.
  3. Line 46. Covalently, not convanlently. BTW, I do not think it is correct that all HCMV envelope glycoprotein complexes are covalently linked.
  4. Line 122. Unique short genes: this should read genes from the unique short (US) region.
  5. Line 200. Function, not fuction

Author Response

Dear Dr. Hornjak,

Thank you for your comments and for those of the reviewers regarding our recently submitted manuscript. We appreciate the time and effort that have gone into reviewing our manuscript and for the constructive comments of the reviewers.

We have modified the paper, taking into account the reviewers’ comments. A point-by-point response is included below.

We hope that the manuscript is now acceptable for publication.

We look forward to your response.

Sincerely,

Pilar Pérez-Romero

REVIEWER 3

The authors of this manuscript did an in-silico analysis of 9 HCMV strains to identify viral proteins with transmembrane (TM) domains. They used the GenBank-annotated sequences of these 9 HCMV and analyzed them with three prediction software packages, PureseqTM, Phobius, and TMHMM. With these methods, 94 proteins were initially predicted to be putative TM proteins. 14 of those were excluded from further analysis because they have been shown not to be TM proteins in published experimental studies. 42 of the remaining 80 proteins were conserved in all 9 strains analyzed and were defined as core TM proteome. The proteins were further subdivided into different categories based on published data and homology to proteins from other organisms or pathogens.

Overall, the study is well presented and easy to read. The classification as a putative transmembrane protein may be helpful for future studies. However, the reader would benefit from information regarding the reliability of these predictions. Unfortunately, none of the proteins newly predicted to be TM proteins have been analyzed in biological experiments. Thus, it remains unclear how many of the predicted TM proteins truly are TM proteins – 14 were already discarded as obvious false positives.

Response: We appreciate the reviewer’s comments and suggestions. The work presented in the manuscript aims to identify new targets with TM domains that may potentially be involved in virus-cell interactions during infection using bioinformatic tools. These proteins will be used for further experimental characterization to describe their role during CMV infection. Although this experimental approach was not part of this study, following the reviewer suggestion we have included, as an example, new data (Figure S3) regarding two of the identified proteins with unknown function (UL2 and UL124). The ORFs of these proteins were cloned into an expression plasmid (that includes a myc tag) and expressed in HEK 293T cells. After transfecting the HEK 293T mammalian cell line, plasma membrane proteins were extracted and the cytoplasmic and plasma membrane protein fractions were tested by Western Blot using an anti Myc antibody. Both UL4 and UL124 proteins were only detected in the PM fractions confirming their location in the membrane. This information was added in the results section.

Specific questions and comments

  1. The authors chose 9 HCMV strains (actually only 8 as AD169 is included twice) for their analyses, but they do not explain why they chose these strains. In fact, this reviewer would argue that the choice of HCMV strains is rather poor. 3 strains have been passaged extensively in cell culture (AD169, AD169-BAC20, Towne) or at least for several passages (Toledo, TB40E, TR). As HCMV is known to accumulate mutations quite rapidly in cell culture, the usefulness of these cell culture-adapted strains is rather limited (PMID 25894764). They encode numerous mutations, deletions, and sometimes even inversions, which will confound the analyses. Hence, the most suitable strains among the 9 are probably Merlin and HANSCTR4, maybe also VR7863. These strains most likely encode a true and complete HCMV proteome. There are more than 200 complete HCMV genomes in GenBank. In some cases, the viral genome sequence was determined directly from clinical material, prior to cell culture passaging. Why did the authors not use such HCMV genomes for their comparative analyses?

Response: The reviewer is right, and in order to test to what level these 9 selected CMV strains are representative of the 335 available CMV genomes in GenBank, the 56,190 ORFs were aligned with the ORFs in our CMV dataset. We obtained 100% median percentage identity and breadth coverage (overlapping distance), representing 99.95% of the total ORFs from the Human betaherpesvirus 5 in the NCBI database. This information has been included in the Results, Materials and methods and in Figure S1A.

  1. Table 1 lists the 9 HCMV strains used in this study and the original sources of the strains. The information as presented is incomplete and therefore misleading. The authors should include in the table information on how extensively these viruses have been passaged in cell culture before the sequence was determined. They should also make the non-expert reader aware of the fact that the extensively passaged HCMV stains are not genetically intact.

Response: The reviewer is right, and following the reviewer´s suggestion we have included in Table 1 the information regarding the number of passages of the viruses in cell culture before sequencing, and we have acknowledge this issue in the results section.

  1. Dis the authors re-analyze / re-annotate the viral genomes or did they rely on the annotations found in the GenBank files? The ORF predictions are often based on simple algorithms. There are several proteomic studies that have used proteomic methods to analyze the protein coding content of HCMV. These studies have shown that some of the annotated ORFs do not code for proteins, whereas other previously not identified ORFs have been identified (e.g., PMID 23180859, 32486127, 31873071). Did the authors use information from these proteomic studies to curate their ORF database? One proteomic study has specifically focused on membrane proteins (PMID 24906157). Did the authors consider the results of this study?

Response: We appreciate the reviewer’s comments and suggestions. We did not reanalyze/reannotate the viral genomes but we analyzed each predicted ORF with Blast in order to find orthologs. Following the reviewer’s suggestion, we have included these references and we have revised our dataset taking into account these studies.

  1. Defining a “core TM proteome” (Fig 3) based on a comparison of 9 strains, several of which are severely mutated, does not seem appropriate. The authors could use BLAST (or a similar software) to determine whether the 80 putative TM proteins are conserved in most HCMV strain sequences present in GenBank. Excluding a protein because it is not present in AD169 or Towne seems totally inappropriate.

Response: We appreciate the reviewer’s comments and suggestions.  Following the reviewer suggestion, as previously stated, the same set of 335 CMV genomes were aligned to our set using blastp under default parameters to all proteins contained in our core pangenome. All genomes had representative proteins related to proteins in our set in different proportions with a high number of genomes containing all 39 core proteins. This information has been included in the Results, Materials and methods and in Figure S1B.

  1. The signal peptides of type I TM proteins are usually quite hydrophobic and are often predicted to be TM domains, which they are not as the signal peptide is not present in the mature protein. Did the authors account for signal peptides in their analyses?

Response: The reviewer´s is right, and following the reviewer’s suggestion, the tool SignalP6.0 was used for signal peptide analysis. As a result, UL4, UL22A and UL116, that were predicted to have one transmembrane domain only with TMHMM method, were discarded because the transmembrane domain corresponded to the sequence of signal peptide. A paragraph including this information was added in the results section and the discarded proteins were added to Figure S2.

Minor points

  1. Figure 4. There are weird characters and symbols in this figure, probably due to use of an incompatible font.
  2. The supplemental figures are in need of legends.
  3. Line 46. Covalently, not convanlently. BTW, I do not think it is correct that all HCMV envelope glycoprotein complexes are covalently linked.
  4. Line 122. Unique short genes: this should read genes from the unique short (US) region.
  5. Line 200. Function, not fuction

Response: Following the reviewer’s suggestion, we have included the suggested changes.

Reviewer 4 Report

I read with very interest the article entitled “Deciphering the Potential Coding of Human Cytomegalovirus: 2 New Predicted Transmembrane Proteome”.

Authors investigated about CMV proteome in order to identify new targets potentially involved in virus-cell interaction during infection, useful for neutralizing antibodies and for the development of novel vaccine.

Overall the study is well-conceived and the manuscript well-written. The topic is really interesting and the methods are clearly illustrated. Findings are intruiguing and well reported. A mention should be done to seminal works in this field, namely regarding the association between γ marker, human leucocyte antigens and killer immunoglobulin-like receptors and the natural course of human cytomegalovirus infection, consider mentioning the following excellent papers PMID: 24973460, PMID: 24737799, PMID: 29067686; PMID: 30764515.

Author Response

Dear Dr. Hornjak,

Thank you for your comments and for those of the reviewers regarding our recently submitted manuscript. We appreciate the time and effort that have gone into reviewing our manuscript and for the constructive comments of the reviewers.

We have modified the paper, taking into account the reviewers’ comments. A point-by-point response is included below.

We hope that the manuscript is now acceptable for publication.

We look forward to your response.

Sincerely,

Pilar Pérez-Romero

Reviewer 4

I read with very interest the article entitled “Deciphering the Potential Coding of Human Cytomegalovirus: 2 New Predicted Transmembrane Proteome”.

Authors investigated about CMV proteome in order to identify new targets potentially involved in virus-cell interaction during infection, useful for neutralizing antibodies and for the development of novel vaccine.

Overall the study is well-conceived and the manuscript well-written. The topic is really interesting and the methods are clearly illustrated. Findings are intruiguing and well reported. A mention should be done to seminal works in this field, namely regarding the association between γ marker, human leucocyte antigens and killer immunoglobulin-like receptors and the natural course of human cytomegalovirus infection, consider mentioning the following excellent papers PMID: 24973460, PMID: 24737799, PMID: 29067686; PMID: 30764515.

Response: We appreciate the reviewer’s comments and suggestions. Following the reviewer’s suggestion, we have included the references and also included a paragraph in the discussion.

Round 2

Reviewer 3 Report

The authors have responded adequately to my comments and concerns. They have made appropriate changes thereby improving the quality of the study.